# Targeting NMDA Receptors in Emotional Disorders: Their Role in Neuroprotection

**DOI:** 10.3390/brainsci12101329

**Published:** 2022-09-30

**Authors:** Siqi Wang, Lihua Bian, Yi Yin, Jianyou Guo

**Affiliations:** 1CAS Key Laboratory of Mental Health, Institute of Psychology, Chinese Academy of Sciences, Beijing 100101, China; 2Department of Psychology, University of Chinese Academy of Sciences, Beijing 100049, China; 3Wenzhou Institute, University of Chinese Academy of Sciences, Wenzhou 325001, China

**Keywords:** NMDAR, depression, anxiety, neuroprotection

## Abstract

Excitatory glutamatergic neurotransmission mediated through N-methyl-D-Aspartate (NMDA) receptors (NMDARs) is essential for synaptic plasticity and neuronal survival. While under pathological states, abnormal NMDAR activation is involved in the occurrence and development of psychiatric disorders, which suggests a directional modulation of NMDAR activity that contributes to the remission and treatment of psychiatric disorders. This review thus focuses on the involvement of NMDARs in the pathophysiological processes of psychiatric mood disorders and analyzes the neuroprotective mechanisms of NMDARs. Firstly, we introduce NMDAR-mediated neural signaling pathways in brain function and mood regulation as well as the pathophysiological mechanisms of NMDARs in emotion-related mental disorders such as anxiety and depression. Then, we provide an in-depth summary of current NMDAR modulators that have the potential to be developed into clinical drugs and their pharmacological research achievements in the treatment of anxiety and depression. Based on these findings, drug-targeting for NMDARs might open up novel territory for the development of therapeutic agents for refractory anxiety and depression.

## 1. Introduction

Mental health is a major public health challenge in the world today. With the development of science and the economy, people are beginning to pay more attention to various mental health problems in social life. Especially during the coronavirus pandemic, home confinement, social isolation and the resulting economic breakdown have motivated the general public to show increased symptoms of depression, anxiety and stress-related to COVID-19 [1,2,3,4,5]. This risk of deterioration in social mental health problems has made mental health issues enter the public view again [6].

According to the analysis for the global burden of disease study [7], neuropsychiatric disorders account for approximately 14% of the global burden of disease—including schizophrenia, depression, anxiety and substance-related disorders. Mental disorders not only increase the risk of infectious and non-infectious diseases, but also cause unintentional or intentional injuries to the body. Accordingly, mental disorders are often accompanied by higher rates of suicide and remain one of the leading causes of disability worldwide. There are over 80 million people dead from suicide each year. Most suicides are related to mental illness and the majority of these have a diagnosis of at least one mental disorder. Among all cases of unnatural deaths globally, depression accounts for 30%, followed by substance-related disorders (18%) and schizophrenia (14%) [8].

However, due to a variety of reasons, such as living environment and social status, more than 70% of people who need mental health services still lack access to effective treatments [9]. At the current stage, the understanding of brain functions and their molecular and cellular mechanisms are still limited. The complex etiology and pathophysiological mechanism of psychiatric disorders and the comorbidities between different psychiatric disorders often lead to the low efficacy of current clinical psychotherapeutic treatments—particularly in medium-to high-severity cases. Meanwhile, pharmacological treatments are influenced by individual differences, which leads to poor specificity, high drug resistance and high recurrence rates [10,11]. Therefore, it is crucial to continually explore the etiopathogenesis of mental disorders and identify potential biomarkers to guide the diagnosis and development of potential new drugs. 

The normal neurological functions of the brain involve many types of neurons and intercellular signaling paths. Glutamate is an important excitatory neurotransmitter in the central nervous system and participates in excitatory conduction in the nervous system. At present, more and more evidence has shown that disorders of glutamatergic neurotransmission can affect normal brain functions and emotional regulation, which is related to the development of a variety of mental diseases [12,13,14]. The N-methyl-D-Aspartate (NMDA) receptor (NMDAR) is one of the most important receptors involved in the neurotransmission of glutamate; it plays a significant role in the neurobiological mechanisms involved in emotion. Furthermore, along with esketamine—a non-competitive NMDAR antagonist that has been approved as a therapy for treatment-resistant depression (TRD) for adults by the FDA [15]—research into NMDARs has received much more attention in recent years. Unlike monoamine neurodrugs, which work by increasing serotonin or norepinephrine, NMDARs are more involved in regulating synapses and neuroplasticity to promote neuron survival. Research also suggests that for emotional disorders, the regulation of synapses and neuroplasticity may be more crucial for their treatment [16,17]. In this review, based on the biological mechanisms of NMDARs in emotion regulation, we analyzed the neuroprotective mechanisms of NMDARs in synaptic plasticity and neuronal survival—focusing on a summary of therapeutic mechanisms and relevant drug targets involving NMDARs in anxiety and depression in animal models and clinical trials, laying the foundation for the further study of NMDARs. 

## 2. NMDA Receptors

NMDAR is an ionotropic glutamate receptor with high permeability to calcium ions (Ca^2+^). It is widely expressed in the central nervous system—especially in the hippocampus, cortex, amygdala and other brain regions. The family of NMDARs consists of three different subfamilies: GluN1, GluN2 and GluN3 [18,19,20]. In addition, the GluN1 subunit is encoded by a single gene and then alternatively spliced into eight distinct isoforms. The GluN2 subunits are divided into four subunits including GluN2A, GluN2B, GluN2C and GluN2D, which are encoded by four different genes. GluN3 subunits are also subdivided into GluN3A and GluN3B, arising from two individual genes. A common functional NMDA receptor is a tetramer complex containing two necessary GluN1A subunits and two GluN2 or GluN3 subunits. GluN1 can also form triheteromeric NMDARs with two different GluN2 subunits in specific neurons [21,22].

NMDAR subunits all have a significant degree of homology and high structural dependance, with four domains: the extracellular amino-terminal domain (ATD), the extracellular ligand-binding domain (LBD), the transmembrane domain (TMD) and an intracellular carboxyl-terminal domain (CTD). The LBD contains the agonist binding site, and the activation of NMDARs requires the simultaneous binding of both glutamate and glycine. The GluN1 and GluN3 subunits possess recognition sites for glycine, while GluN2 subunits carry recognition sites for glutamate [20]. Mg^2+^ ions are removed through cell membrane depolarization, and the co-agonists glutamate and glycine bind to the ligand part—thus achieving the activation of NMDARs. 

The various combinations of NMDA homologous subunits also contribute to the different types of NMDAR subunits in cells at different stages of the central nervous system. The GluN1 subunit is an essential subunit of functional NMDARs, which is widely present in central neurons [23]. In rodents, GluN2B and GluN2D exist widely throughout the embryonic brain, while GluN2B is highly expressed only in the forebrain of adults. The expression of GluN2A gradually increases beginning from birth. With the abundant expression of GluN2A in the brain, the expression of GluN2D decreases gradually until it is slightly expressed in the adult midbrain and diencephalon. GluN2C is expressed in the cerebellum and olfactory bulb two weeks after birth [24,25]. GluN3A, on the other hand, reaches its peak within two weeks after birth, then plummets and maintains a low level in the brain in adulthood [26,27]. GluN3B increases gradually during development and mainly exists in motor neurons in adulthood [28]. Previous findings have confirmed that different stages of neuronal development require the specific expression of various subunits; mutations in genes encoding these subunits also lead to abnormalities of brain functions, affecting the normal neurological function of the organism. *GRIN2A* and *GRIN2B* genes encode the GluN2A and GluN2B subunits and are most frequently associated with diseases related to NMDARs. *GRIN2A* mutations have always been shown in epilepsy-aphasia spectrum disorders [29]. In addition, mutations in *GRIN2B* lead to synaptic dysfunction [30]. What is discussed above has demonstrated the importance of NMDAR subunits in neuronal function and neural development.

## 3. NMDAR-Mediated Signaling in Brain Functioning

### 3.1. Synaptic Plasticity

Synaptic plasticity is an essential feature of neurons that maintains normal brain behaviors, learning and memory emotions; it specifically refers to the activity-dependent modification of the strength or efficacy of synaptic transmission at pre-existing synapses [31]. In the nervous system, Ca^2+^ is a ubiquitous secondary messenger that controls the excitability of neurons, the development of neuronal morphology, the formation of synapses and synaptic plasticity [32]. NMDARs have a distinctly high Ca^2+^ permeability, which helps them to receive excitatory neurotransmitter glutamate delivered from the presynaptic membrane, activate NMDA-dependent ion channels and induce Ca^2+^ influx. Combined with their unique characteristics of Mg^2+^ blockade at resting potential and slow ligand-gated kinetics, NMDARs can also play an important role in the regulation of synaptogenesis, synaptic remodeling and synaptic plasticity [33,34]. NMDAR-dependent Ca^2+^ influx can induce long-term potentiation (LTP) and long-term depression (LTD), which are crucial forms of synaptic plasticity in the central nervous system (Figure 1). Nevertheless, abnormal LTP and LTD in the brain are often the pathological mechanism of various brain diseases such as depression, anxiety, addiction and schizophrenia—which indicates that NMDARs are involved in the pathophysiological processes of various diseases via the regulation of LTP and LTD.

The most typical NMDAR-dependent LTP occurs mainly between the CA3 and CA1 pyramidal neurons of the hippocampus. In LTP, the intracellular Ca^2+^ concentration increases when NMDARs are activated—triggering the activation of calmodulin-dependent protein kinase II (CaMKII), which induces the phosphorylation of downstream signaling proteins. Then, this leads to the profound reorganization of the molecular composition of the postsynaptic density, subsequently inducing the enlargement of dendritic spines and increasing the number of AMPARs on the plasma membrane—thereby producing enhanced synaptic strength [35]. Studies have confirmed that protein kinase C (PKC) can potentiate NMDA currents and modulate NMDAR gating properties; additionally, it also improve the NMDA channel-opening rate via SNARE-dependent exocytosis [36,37,38]. The Ca^2+^ permeability of NMDARs can be directly activated by protein kinase A (PKA), promoting Ca^2+^ signaling in dendritic spines and inducing LTP at the hippocampal CA1 region [39,40]. 

It is generally believed that more the intense activation of NMDARs and postsynaptic threshold-exceeding Ca^2+^ concentrations are the key to triggering LTP, while the induction of NMDAR-dependent LTD only requires moderate activation of NMDARs [41]. Unlike LTP, LTD is mainly induced by appropriate Ca^2+^ levels, which more readily binds with CaMKII and activates protein phosphatase 1 (PP1)—leading to the dephosphorylation of AMPAR subunits or downstream kinases and promoting the endocytosis of postsynaptic AMPARs [42,43]. NMDAR-dependent LTD may be closely related to the reduced number of surface AMPARs at the synapse and on dendrites. Recent studies have indicated that LTD induced by NMDARs is accompanied by the activation of p38 MAPK. Meanwhile, it has been confirmed that the activation of p38 MAPK is necessary for NMDAR-dependent LTD to drive spinal dendrite spinous contraction and synaptic weakening, which is independent of signals passing through AMPARs [44,45,46]. Moreover, another study has also proposed that activating Bax-related apoptotic mechanisms via mitochondria is required for NMDAR-dependent LTD [47].

As the research into NMDAR subunits further develops, additional evidence has come to suggest that the differential activation of NMDAR subunits also factors in determining which type of plasticity is induced—suggesting that NMDAR-mediated LTP and LTD are dependent on the activation of specific NMDAR subtypes. The induction of LTP involves the activation of primarily triheteromeric NMDARs containing both the GluN2A and GluN2B subunits [48,49], while neither the GluN2A nor the GluN2B subunit is strictly necessary for NMDAR-dependent LTD [50].

### 3.2. Neuronal Survival

In addition to playing a crucial role in synaptic transmission and synaptic plasticity, recent studies have also found that activation of NMDARs can promote the survival of neurons under certain physiopathological conditions. Synaptic NMDAR-dependent neuronal survival mainly includes the induction of survival genes, the suppression of death genes and protection against oxidative stress (Figure 1) [51,52]. Dependent on the elevated concentrations of intracellular Ca^2+^ following NMDAR activation, Ca^2+^ can be transduced to the nucleus, which activates the extracellular signal-regulated kinase cAMP response element-binding protein (CREB) through the nuclear Ca^2+^ signaling pathway and regulates the expression of downstream brain-derived neurotrophic factor (BDNF) to promote neuronal survival, growth and differentiation [53,54]. By activating CREB, NMDARs can promote the expression of transcription repressors for death-signaling genes such as transcription factor 3 (ATF3), causing ATF3-mediated transcriptional repression and inhibiting cell death [55]. NMDARs can also exert neuroprotective effects through PI3K/AKT and the MAPK/ERK pro-survival signaling pathway [56,57]. Studies have found that activation of synaptic NMDARs can inhibit the expression of the pro-apoptotic gene Puma and nuclear translocation of the death transcription factor forkhead box O (FOXO) [52,58]. In addition to their cell survival-promoting and apoptosis-suppressing effects, synaptic NMDARs can enhance cellular antioxidant defenses and protect cells from oxidative stress-induced neuronal death [59]. 

On the contrary, an article in the early 2000s demonstrated for the first time that overactivated NMDARs under abnormal conditions bring about the dephosphorylation of CREB, leading to its inactivation [60]. Subsequent studies have confirmed that synaptic and extrasynaptic NMDARs stimulation initiates dramatically different physiological changes. The activation of extrasynaptic NMDARs triggers the cell death pathway, which is mostly associated with the closure of the relevant pro-neuronal survival pathway and the destruction of mitochondrial function [61,62]. Activated extrasynaptic NMDARs antagonize nuclear signaling to CREB, which blocks BDNF expression and participates in mitochondrial dysfunction to induce apoptosis. At the same time, stimulation of the extrasynaptic NMDARs does not induce the activation of ERK, but evokes ERK inactivation and inhibits neuronal survival [62]. Activation of extrasynaptic NMDARs also promotes the nuclear translocation of FoxO3a, an inducer of cell death-promoting genes, to induce cell death [58]. By comparison, the transcriptional coactivator PGC-1α can antagonize neuronal apoptosis caused by extrasynaptic NMDARs, contributing to neuroprotective effects [63].

In line with the regulation of synaptic plasticity mediated by NMDARs, the current prevailing theory suggests that the activation of low-dose agonists can preferentially activate NMDARs in the synapse and stimulate CREB, ERK and other molecular targets—thereby activating related downstream pro-survival signaling. High-dose agonists act on both synaptic and extrasynaptic NMDARs, and the activation of extrasynaptic NMDARs counteracts the activation of synaptic NMDARs—suppressing pro-survival signaling and leading to neuronal death [62]. However, recent studies have confirmed that activation of synaptic or extrasynaptic NMDARs alone does not lead to NMDAR-dependent cell death, but rather activates pro-survival signaling. Only the co-activation of synaptic and extrasynaptic NMDARs can induce NMDAR dependent-cell death programs [64,65], and the degree of activation is related to the magnitude and duration of extrasynaptic and synaptic NMDAR co-activation [65]. The drug memantine plays a neuroprotective role by blocking intracellular pro-death signaling mediated by the simultaneous activation of synaptic and extrasynaptic NMDARs. The multiple different combinations of the seven subunits of NMDARs also indicates the complex physiopathological properties of NMDARs. A good overview of the pharmacological and structural differences between synaptic and extrasynaptic NMDARs will facilitate the development of more targeted drugs.

## 4. Role of NMDA Receptors in Anxiety

Nowadays, anxiety is one of the most common mental disorders and is also is a widespread symptom of many mental disorders. It is mainly manifested as unconditioned anxiety and conditioned fear. The amygdala, cortex, bed nucleus of stria terminalis (BNST) and ventral hippocampus are all important brain regions involved in the onset of anxiety. The amygdala and its efferent nerves are involved in the acquisition, consolidation and expression of conditioned fear. The functional integrity of the prefrontal cortex (mPFC) has the ability to resist anxiety [66]. The BNST is a vital forebrain region for the integration of stress and anxiety responses [67] and the ventral hippocampus is necessary for hippocampus-dependent fear memory. Stress is a crucial risk factor for the development of anxiety disorders and exposure to stress may alter glutamatergic neurotransmission in brain regions associated with anxiety. Acute stress increases glutamatergic transmission in the mPFC and hippocampus. Anxiety-like phenotypes tend to be accompanied by decreased activity in the mPFC and increased activity in the basolateral amygdala (BLA) [68,69]. Disruption of BNST excitability also induces an increase in anxiety-like behaviors [70,71]. 

Numerous studies have confirmed that glutamatergic neurotransmission plays a significant role in the pathogenesis of anxiety disorders, which suggest that NMDARs may be important in regulating anxiety-like behaviors [72,73]. However, the pathogenesis of anxiety is more complex, and the current research on NMDARs in anxiety also requires further investigation. Hippocampal suppression of GluN2A- or GluN2B-containing NMDARs inhibits anxiety-like behaviors [74]. Inducing inactivation of the NMDAR GluN2A subunit has also shown antianxiety effects in a variety of animal experiments [75]. It has also been shown that the expression of GluN2B subunits is decreased in the mPFC and hippocampus but increased in the amygdala in anxious rats [76]. Mice with knockdown of the GluN2D subunit gene in BNST have deepened negative emotions and increased anxiety-like behaviors [70]. Moreover, activation of NMDAR-dependent LTP in the BNST promotes anxiolytic effects [71]. When the GluN2C subunit gene is replaced by the GluN2B subunit gene over the whole mouse brain, a significant unconditioned anxiolytic behavior has been observed in 1-month-old mice [77]. Due to the key role of NMDARs in synaptic plasticity and neuronal survival, the normal development of the early brain is inseparable from NMDARs. During this period, NMDAR hypofunctioning is highly susceptible to inducing apoptosis of the cerebral cortical and limbic system, which has irrecoverable effects on the body [78]. One study found that male neonatal mice treatment with phencyclidine (PCP), a NMDAR antagonist, can induce elevated anxiety-like levels in adulthood [79]. Studies have demonstrated that prenatal stress can also induce a decrease in GluN1 and GluN2A expression in the brain of mouse offspring, promoting animals to exhibit anxiety-like behaviors [80]. All the above studies confirm that there are significant differences in NMDARs involved in the regulation of anxiety-like behaviors across time, brain regions and subunit types—suggesting that both NMDAR antagonists and partial agonists have anti-anxiety effects to a certain extent. 

Heretofore, benzodiazepines and selective serotonin uptake inhibitors (SSRIs) have been widely used in the clinical treatment of anxiety disorders, which respectively target γ-aminobutyric acid (GABA) and monoaminergic neurotransmission. Although widely used, these drugs still have common psychotropic side effects, including weight loss, gastrointestinal discomfort and sedation. Moreover, these drugs are not effective for every patient. With the understanding of NMDARs from abundant clinical and preclinical studies, the application of NMDAR-related drugs in the treatment of anxiety disorders has been recommended recently. There are NMDAR modulators in clinical trials for anxiety, as shown in Table 1. D-cycloserine (DCS) is a partial agonist at the glycine recognition site of the NMDA receptor in the amygdala. Previously applied in the chronic treatment of human tuberculosis, it has now been clinically found to be an adjunctive agent that enhances the anti-anxiety efficacy of psychotherapy [81,82]. Experimentally, patients with panic disorder were used, and were gave a single oral dose of DCS along with the psychological treatment of single-session exposure therapy. Via the statistical analysis of a face dot-probe task, magnetic resonance imaging (MRI) and clinical scales, the trial found that patients with DCS administration showed significantly lower responses to fearful faces and aversive images, and showed clearly greater clinical recovery rates at 1-month follow-up [82]. DCS administration in this study also showed no serious adverse events, suggesting the clinical feasibility of DCS as an adjuvant agent to augment psychological treatments. Glycine/D-serine has been proven to act as a primary endogenous co-agonist of glutamate for the activation of NMDARs. However, there is substantial evidence from rodent models relevant to anxiety that the direct NMDAR agonists glycine and d-serine have prominent anxiogenic-like effects in mice in multiple tests of anxiety and that glycine transporter-1 (GlyT-1) inhibitors such as ALX-5407 could also cause anxiety in mice [83], while in mutant mice with reduced NMDA-NR1 glycine affinity, they demonstrated an anxiolytic-like phenotype [83]. D-amino acid oxidase (DAO, DAAO) is an enzyme that can degrade d-serine. D-amino acid oxidase knockout mice show enhanced short-term memory performance and the extinction of contextual fear memory, but also heightened anxiety [83,84,85]. These findings suggest that therapies targeting the NMDAR glycine site may require further investigation to ensure that there is no induction or exacerbation of anxiety syndromes. In addition to DCS, ketamine is also used clinically to treat refractory anxiety disorders. Ketamine has been a hot research topic for non-competitive NMDAR antagonists in recent years. Numerous case reports have shown that ketamine can produce rapid anti-anxiety effects in patients suffering from refractory generalized anxiety or social anxiety disorders [86,87,88], and a single injection of ketamine can provide sustained relief from multiple anxiety symptoms in patients [87]. Although the anti-anxiety effect of ketamine only lasts for almost two weeks and its specific mechanism of action is not yet clear, its pharmacokinetic differences from traditional anti-anxiety drugs that require long-term administration for efficacy also provides more possibilities for clinical anti-anxiety treatments [89]. 

In addition, there have been plentiful preclinical studies on the anxiolytic effects of other agonists or inhibitors of NMDARs. For example, fixed-site injection of the NMDAR competitive antagonist AP5 into the amygdala, middle nucleus, ventral hippocampus—or all of those areas—reduces anxiety-like symptoms in rodents [72]. MK-801, the non-competitive NMDAR antagonist, also shows anti-anxiety effects in elevated cross maze tests in rats [90]. PCP has an anxiolytic effect in rodent anxiety models [72]. Some studies have also revealed the anti-anxiety effects of Chinese herbal drugs via NMDARs [91,92]. Lavender essential oil and its main components exert an affinity for glutamate NMDARs in a dose-dependent manner at certain components, indicating that its relaxing, soothing and anti-anxiety effects might be related to the modulation of NMDARs [91]. Diosgenin, a steroidal saponin that is widely found in legumes, could alleviate anxiety-like behaviors in young mice by increasing the expression of NMDARs in the hippocampus [92]. Thus, all of these studies have provided evidence for the impact of NMDARs in reducing anxiety, suggesting that NMDAR modulators can be involved in mood regulation and exert neuroprotective effects.

**Table 1 brainsci-12-01329-t001:** Novel NMDARs modulators in clinical trials for anxiety.

Compound	Mechanism of Action	Effect
D-cycloserine (DCS)	NMDAR glycine partial agonist	Augmenting extinction learning [93]; adjunct treatment to exposure therapy for anxiety disorders [82]
Ketamine	Non-competitive NMDA channel blocker	Reducing anxiety in treatment-resistant generalized anxiety and social anxiety disorders [86,94]
Memantine	Non-competitive NMDAR antagonist	Treatment-resistant obsessive-compulsive disorder [95]
Amantadine	NMDAR antagonist	Augmentation therapy for obsessive-compulsive patients resistant to SSRIs [96]

## 5. Role of NMDA Receptors in Depression

Depression is a prevalent mental disorder and a growing public health problem in society today. It is characterized by low mood, anhedonia and hopelessness, along with other changes such as weight changes, difficulty sleeping and retardation of thinking. Major depression disorder (MDD) is associated with high rates of suicide and is a primary cause of disability worldwide. However, as a complicated mental disease, the pathophysiological causes of depression remain unclear. Some patients usually show depression-like symptoms and other psychotic symptoms. Moreover, multimorbidity (the presence of two or more chronic conditions) can aggravate the risk of depression [97,98]. Early studies on the pathogenesis of depression mainly focused on the monoamine hypothesis and neurotrophic factor hypothesis. Negative emotions are caused by under-expressed serotonin and dopamine, while decreased levels of neurotrophic factors in the brain contribute to brain damage. However, this alone does not explain the onset of depression. In neuroanatomical studies of patients with MDD, it was found that patients exhibit significant brain volume reduction, with lower cortical thickness and cell densities in the PFC [99]. Meanwhile, decreased synaptic connections and the loss of excitatory synapses could be observed in the PFC and hippocampus [100,101]. These results indicate that stress and depression involve neuronal atrophy and synaptic loss in several brain regions such as the cortex and hippocampus. Locus coeruleus (LC) and amygdala neuronal activity receive glutamatergic input from PFC neurons and participate in brain emotion regulation. It has also been shown that abnormal neurogenesis and synaptic activity is present in the BLA from postmortem brain analyses and imaging studies of patients with depression, indicating the dysfunction of local neuroplasticity [102,103]. These studies suggest that stress can induce abnormal synaptic structure and function in different brain regions, which results in impaired neuroplasticity and functional connections in brain regions. 

As a result of the last two decades of research targeting the glutamatergic system combined with the functional regulation of NMDARs in synaptic plasticity and neural survival, a connection between NMDAR dysfunction and depression has also been suggested. Firstly, abnormal expression of NMDAR-related genes has been observed in patients with depression [104,105]. Autopsy results from MDD patients have shown decreased levels of both GluN2A and GluN2B subunits in the PFC [106], increased expression of the GluN2A subunit in the lateral amygdala [107], and elevated levels of the GluN2B and GluN2C subunits in the LC [108,109]. Moreover, compared with non-suicidal patients with MDD, the expression of the human gene *GRIN2B*, encoding the NMDAR GluN2B subunit in the brain of suicidal depression patients, is increased—suggesting that *GRIN2B* could be a candidate gene for susceptibility to MDD [105,110]. Meanwhile, a great number of preclinical studies targeting NMDAR subunits in depression models have also confirmed that stress could induce the abnormal expression of NMDARs. In the rat learned-helplessness model of depression, higher concentrations of the GluN2B subunit can be observed in both the postsynaptic density and the postsynaptic cytoplasm [111]. Additionally, maternal separation stress increases the gene expression of the GluN2A (but not the GluN2B) subunit of NMDARs in the hippocampus [112]. Under chronic unpredictable stress and chronic restraint stress, male rats show a decrease in GluN1 and an increase in GluN2B in the hippocampus [113,114]. It has also been suggested that inactivation of the GluN2A subunit and downregulation of GluN2B subunit expression can evoke antidepressant-like activity in mice [75,115]. Mice lacking the GluN2A subunit do not exhibit depression-like behavior after LPS treatment, which indicates an important regulatory role of the GluN2A subunit in neuroinflammation-related depression [116] In line with the complex regulatory mechanisms of NMDARs in anxiety studies, antidepressant studies involving NMDARs also have shown contradictory conclusions from different subunits in different brain regions. This may be related to the type of stressors, the duration of stress and the differing functions of different brain regions. These studies indicate that targeted NMDAR drugs may serve as a new direction for the development of antidepressant drugs. There are NMDARs modulators in clinical trials for depression, as shown in Table 2.

Currently, the most widely used antidepressants are first-generation antidepressants, including monoamine oxidase inhibitors (MAOIs) and tricyclic antidepressants (TCAs), and second-generation antidepressants such as selective serotonin reuptake inhibitors (SSRIs) and serotonin-norepinephrine reuptake inhibitors (SNRIs)—mainly based on the early monoamine theory of depression. Although these drugs have shown antidepressant effects, there are still approximately one-third of patients who are not sensitive to these classical antidepressants. Additionally, these drugs cannot alleviate suicidal behavior in patients with MDD. With the first study of NMDAR antagonists in depression-like animal models in the 1990s, it was revealed that both the competitive antagonist AP7 and the non-competitive antagonist MK-801 could reduce the behavioral despair duration of animals in forced swimming and tail suspension experiments, showing antidepressant effects [117]. The discovery of the rapid antidepressant effects of ketamine could be considered a major breakthrough in current depression research. Ketamine was originally a clinical dissociative anesthetic, consisting of equal amounts of (S)-ketamine (or esketamine) and (R)-ketamine. In the first clinical double-blind study of ketamine in antidepressant treatment, it was found that a single intravenous injection of ketamine hydrochloride with a dose of 0.5 mg/kg over 40 min begins to exert an antidepressant effect within 4 h after infusion—peaking at about 72 h—and that this single treatment could be sustained in patients for a week or longer [118,119]. Intravenous administration of ketamine twice or three times a week could also maintain antidepressant efficacy for more than 15 days in treated patients [120,121]. Studies on the rapid antidepressant mechanism of ketamine suggested that it mainly involves direct NMDAR inhibition, which includes extra-synaptic NMDAR inhibition, inhibition of spontaneous NMDAR-mediated neurotransmission, inhibition of NMDAR-dependent burst firing of lateral habenula neurons and inhibition of GABAergic interneuron NMDARs [122]. Blocking NMDARs on GABAergic inhibitory interneurons could induce the disinhibition of pyramidal neurons and enhance glutamatergic firing. All of the ketamine antidepressant mechanisms described above involve acute changes in synaptic plasticity, leading to the sustained strengthening of excitatory synapses. Additionally, these mechanisms are not contradictory to each other, suggesting that these pathways could synergistically promote rapid antidepressant effects in drugs. At present, esketamine—the S(^+^) enantiomer of ketamine—has been approved for clinical use in depressed patients who are unresponsive to two or more antidepressants [15]. Compared to ketamine, esketamine has a higher affinity for NMDARs and may be safer than ketamine in clinical evaluations [123]. (R)-ketamine may have a more potent and long-lasting antidepressant potential than esketamine and ketamine and appears to be associated with fewer side effects and tendencies for abuse [124]. More detailed clinical safety and efficacy studies for (R)-ketamine are still undergoing research. Although the antidepressant effects in enantiomers of ketamine have been demonstrated, more in-depth research into their underlying mechanisms of action has opened new directions for the development of a new class of safe, rapid and effective antidepressants. 

Dextromethorphan, an NMDAR antagonist, has similar pharmacological characteristics to ketamine—suggesting that it could be studied as a potential antidepressant [125]. Esmethadone (Rel-1017), a noncompetitive NMDAR antagonist, increased BDNF levels in healthy subjects in clinical trials [126]. Furthermore, oral administration of Rel-1017 did not cause psychotomimetic symptoms in patients in a randomized double-blinded phase II trial—suggesting that REL-1017 treatment could provide safe, rapid and long-lasting antidepressant effects for patients with inadequate responses to antidepressant therapies [127]. AZD6765, a low-trapping NMDA channel blocker, has also demonstrated rapid antidepressant effects in early clinical studies—but this significant antidepressant effect was only briefly present for about two hours [128]. Nitrous oxide, a non-competitive NMDAR antagonist, has also shown rapid antidepressant effects in patients with treatment-resistant depression, and these effects could last for a whole day; however, some patients experienced nausea, vomiting and other adverse reactions [129]. DCS-assisted treatment also showed good antidepressant effects in a double-blind placebo-controlled trial of refractory depression patients [130]. GLYX-13 is a novel NMDAR glycine-site functional partial agonist that shows antidepressant-like effects without ketamine-like side effects [131,132]. By giving a single intravenous dose of GLYX-13 in patients with MDD, GLYX-13 reduced depressive symptoms within 2 h, and this effect was maintained for 7 days on average in subjects with MDD who had not responded other antidepressant agents during their current depressive episode [133]. All of these NMDAR modulators have shown certain antidepressant effects. Further understanding of the mechanisms of action of these compounds and the assessment of their safety and efficacy will make the development of antidepressant drugs more comprehensive in the future. 

Antidepressant efficacy studies targeting natural herbal medicines are also a much-researched field in antidepressant drug development at present. Previous experiments in our lab have confirmed that a traditional Chinese medicine formula, the Gan Mai Da Zao (GMDZ) decoction, can regulate glutamate levels and NMDAR subunits in the frontal cortex and hippocampus and then ameliorate depression-like behaviors in sucrose preference and open field tests on chronic stress rats [134]. The extract of radix polygalae (the root of Polygala tenuifolia) antagonizes the activity of NMDARs in the hippocampus and induces a rapid antidepressant effect [135]. YY-23, a modified metabolite of timosaponin B-III, can also exert NMDAR antagonism by directly acting on NMDARs—thus inducing rapid anti-depression effects in the body [136]. Yueju is a traditional herbal medicine formulated to treat depression-related syndromes that has been used for 800 years, inducing a long-term antidepressant effect by regulating GluN1 subunit levels [137].

**Table 2 brainsci-12-01329-t002:** Novel NMDARs modulators in clinical trials for depression.

Compound	Mechanism of Action	Effect
REL-1017 (Esmethadone)	NMDAR channel blocker	Adjunctive treatment in MDD [127]
Ketamine	Non-competitive NMDA channel blocker	Antidepressant mechanisms in TRD and SSRI-resistant depression [121,138,139]
Esketamine	Non-competitive NMDA channel blocker	Antidepressant mechanisms in TRD [15]
Dextro-methorphan	Non-competitive NMDAR antagonist	Novel antidepressant in treatment-resistant MDD [140]
AZD6765	Low-trapping NMDA antagonist	Rapid antidepressant effects in MDD [128]
Nitrous oxide	NMDAR antagonist	Adjunctive therapy in MDD [141]
D-cycloserine (DCS)	NMDAR glycine partial agonist	Adjuvant therapy for treatment-resistant MDD [130,142]
L-4-chlorokynurenine (4-Cl-KYN)	glycine site NMDAR antagonist	Rapid antidepressant effects in TRD [143]
GLYX-13	NMDAR glycine site functional partial agonist	Antidepressant treatment in MDD [131,133]
Traxoprodil (CP-101,606)	NR2B subunit-selective NMDAR antagonist	Antidepressant effects in treatment-resistant MDD [144]

## 6. Discussion

Abundant evidence has proven the role of NMDARs in mediating synaptic plasticity and neuronal survival, and numerous studies have also suggested that synaptic dysfunction and neuroplasticity disorders in the brain are involved in pathological processes related to various mood disorders. NMDARs also play a significant role in the pathophysiology of mood disorders due to their extensive presence in the brain and their various subunit combinations that are involved in the regulation of different physiological signals. Therefore, an in-depth intensive investigation of the NMDAR-dependent synaptic and cellular mechanisms that contribute to mood disorders and the identification of the role of these mechanisms in mood improvement are important challenges that could benefit the development of NMDAR modulators for clinical treatments in today’s world. We have mainly focused on two common emotional disorders—anxiety and depression—and have summarized the results of NMDAR modulators in clinical trials and preclinical studies, finding that they have rapid and robust anti-anxiety and antidepressant effects. The unique neuroprotective effects of NMDAR modulators may eliminate the drawbacks of commonly used drugs for the clinical treatment of mood disorders. The discovery of natural drugs that can act on the NMDAR has also helped to enrich the pharmacology of NMDARs. However, in the process of clinical research, it is also necessary to consider whether NMDAR modulators induce other mental disorders when they function though the modulation of the activity of NMDARs; for example, whether non-competitive NMDAR antagonists could cause NMDAR hypofunction when patients got treatment and induce behaviors of symptoms of schizophrenia such as hallucinations and delusions. Drugs targeting NMDARs could open up a promising new frontier for future research to meet the needs of patients with major anxiety and depression, as well as patients with treatment-resistant depression.

Aside from the anxiety and depression mentioned above, NMDAR modulators also have great potential in the treatment of other psychiatric disorders. Schizophrenia has a high prevalence in the general population, with clinical characters of affective and neurocognitive impairments. Ketamine and PCP, which were developed as dissociative anesthetics, can induce schizophrenic-like symptoms in healthy humans—demonstrating that these effects are mediated via noncompetitive antagonism of NMDARs [145]. All of these suggest that NMDAR dysfunction is a potential schizophrenic causative. The glutamate hypothesis in schizophrenia also focuses on NMDAR hypofunction. Recent studies on endogenous NMDAR modulators in schizophrenia have supported NMDAR deficits in fast-firing GABAergic interneurons and the general attenuation of NMDAR-dependent neuroplasticity at glutamatergic synapses as major research directions in schizophrenia, confirming the important role of the precise regulation of NMDAR function in schizophrenia [146]. In conclusions, understanding the pathophysiological changes of NMDAR hypofunction and overactivation, and deep investigation into the neuroprotection of NMDARs, also provide greater therapeutic windows for the application of NMDAR modulators in other mental disorders such as stroke, Huntington’s and Alzheimer’s disease.

## Figures and Tables

**Figure 1 brainsci-12-01329-f001:**
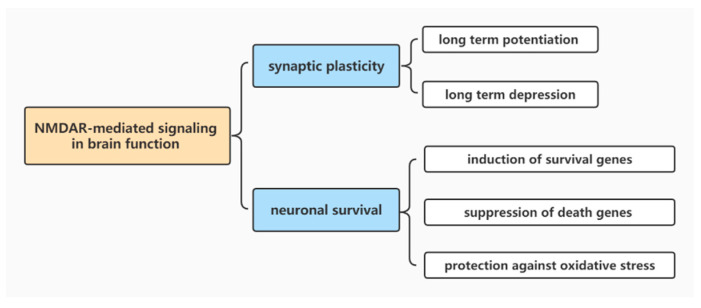
NMDAR-mediated signaling in brain functioning.

## Data Availability

Not applicable.

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
