# Peer review of "Targeting NMDA Receptors in Emotional Disorders: Their Role in Neuroprotection"

_brainsci, 2022, doi:10.3390/brainsci12101329_

Round 1

Reviewer 1 Report

The Wang et al. manuscript is an expertly written, concise , and up to date review on the potential for novel NMDAR modulators to treat depression. Their discussion of neuroprotection was also valuable: now that NMDAR drugs are getting FDA approved for depression, it may be a good time to return to stroke which has been mostly abandoned by pharma with NMDAR drugs since the failure of MK-801. 

Author Response

We appreciate the reviewer’s comments. Please see the attachment.

Reviewer 2 Report

This review aims to compile the scientific data available to date concerning the role of N-Methyl-D-Aspartate (NMDA) subtype of glutamate receptors in the physiopathology of emotional disorders such as anxiety and depression. 

After a concise but well detailed presentation of the structure and functions of this glutamatergic receptor and its subunits, its possible implications specifically in these neuropsychiatric disorders are considered in view of the results from clinical and preclinical studies. In addition, current treatments and therapeutic avenues are presented by which the modulation of NMDA receptors could be seen as a priority strategy in view of the improvements observed and reduced side effects.

A first criticism that I will make to this manuscript is to have poorly considered an important role that could play the glycine site of the NMDA receptor and of the endogenous agonist D-serine. It is known that activation of this site by the amino acid is essential for the proper functioning of the receptors, damage to which is currently implicated in a large number of neuropsychiatric or neurodegenerative diseases. Of course, the manuscript mentions the effects of agonists such as D-cycloserine, but the different studies focusing on D-serine or using mice with a genetic deletion of the serine racemase producing the amino acid should be considered.

A second criticism is based on the fact that the role of NMDA receptors in neuronal survival is considered in some detail (paragraph 3.2) without this aspect being really addressed in the rest of the manuscript. This could be understandable because anxiety and depression are not yet classified as neurodegenerative diseases. I therefore wonder about the relevance of this paragraph.

Minor points:

line 124: NMDA receptors do not necessarily have a unique permeability to calcium, some AMPA receptors can also have it.

lines 148-160: as usually reported in reviews of synaptic plasticity, only the hypothesis of a weak calcium influx through the NMDA receptors is considered for the induction of LTD. However, there is also a metabotropic hypothesis in which it is postulated that it would be a conformational change of the receptor, and in particular of the GluN2B subunits, which would trigger low calcium increase and activation of the phosphatase-related pathways. This is to be mentioned.

Author Response

(The authors gave the same response as above.)

Reviewer 3 Report

The authors provide a comprehensive and well-written description of the NMDA receptors. On the other hand, since it is kind of difficult to read, it would better to list and summarize the recent results of clinical trials targeting this receptor in Table. Also, the author might want to add a few more figures to make it visually easier to read.

Author Response

(The authors gave the same response as above.)
